# Resinous included phloem as a key indicator of authentic or fake agarwood

Jian Feng[1], Yangyang Liu[1]*, Peiwei Liu[1], Yun Yang[1], Anzhen Xie[1], Jianhe Wei[1,2]*

**1** Hainan Provincial Key Laboratory of Resources Conservation and Development of Southern Medicine & International Joint Research Center for Quality of Traditional Chinese Medicine & Key Laboratory of State Administration of Traditional Chinese Medicine for Agarwood Sustainable Utilization, Hainan Branch of the Institute of Medicinal Plant Development, Chinese Academy of Medical Sciences and Peking Union Medical College, Haikou, China, **2** Key Laboratory of Bioactive Substances and Resources Utilization of Chinese Herbal Medicine, Ministry of Education & National Engineering Laboratory for Breeding of Endangered Medicinal Materials, Institute of Medicinal Plant Development, Chinese Academy of Medical Sciences and Peking Union Medical College, Beijing, China

* yyliu@implad.ac.cn (YL); wjianh@263.net (JW)

**Data Availability Statement:** All relevant data are within the manuscript and its Supporting Information files.

**Funding:** Hainan Provincial Natural Science Foundation (822MS145); the project of medical

## Abstract

Agarwood, the resinous wood of the genus *Aquilaria* Lam. and *Gyrinops* Gaertn. of the family Thymelaeaceae, is a very valuable traditional medicinal material and spice. Due to the high economic value of agarwood, unscrupulous merchants have led to the prevalence of fake agarwood in the trade market in pursuit of high profits. Therefore, it is crucial to explore the characteristics of agarwood and establish a simple and rapid method to identify authenticity. This study compared the authenticity of the microstructure of agarwood from different producing areas with that of artificially simulated fake agarwood. The included phloem of authentic agarwood contains brown to brownish resin. And these resins were naturally and stably distributed. The fake agarwood is divided into two categories: one itself is non-agarwood wood (without included phloem), and the other is artificially counterfeiting. However, the filling of resin in the included phloem of agarwood did not occur after artificially simulated counterfeiting treatments. Of the 18 commercially available samples, 10 had the same microstructure as agarwood, but six of them were not completely filled with resin in the included phloem. Therefore, the resinous included phloem is a key characteristic structure of agarwood. It can be used as a basis for authenticity identification of agarwood. This will provide a convenient and rapid method for promoting and popularizing agarwood authentication in trade, customs enforcement, CITES management, and other fields.

## Introduction

Agarwood is the resinous wood of the genera *Aquilaria* Lam. and *Gyrinops* Gaertn. of the family Thymelaeaceae. Its natural distribution range includes China, India, Laos, Cambodia, Thailand, Vietnam, Bangladesh, Indonesia, and Papua New Guinea [1–3]. Agarwood has different names in different cultures of the world, such as 'Chēngxiáng' in China, 'gaharu' in Indonesia and Malaysia, 'agar' in India, 'oud' in the Middle East, and 'jin-koh' in Japan [4, 5]. Agarwood

and health science and technology innovation engineering of Chinese Academy of Medical Sciences (2021-1-I2M-032);Hainan Provincial Nanhai xinxing Science and Technology Innovation Talent Platform Project (NHXXRCXM202341); Hainan Provincial Key Research and Development Project (ZDYF2021SHFZ047); National Natural Science Foundation of China (Grant No.81703660); the earmarked fund for CARS (CARS-21). The funders had no role in study design, data collection and analysis, decision to publish, or preparation of the manuscript.

**Competing interests:** The authors have declared that no competing interests exist.

has been used as a traditional medicine for nearly two thousand years [1] to treat joint pain and is an anti-inflammatory, cardioprotective, sedative, and anticancer agent [6–8]. Agarwood is also a valuable spice with a wide range of applications in fragrances and religion. For example, the volatile oil of agarwood is used as a scent fixative in top-quality perfumes, smoked using agarwood such as agarwood threads, and processed into artifacts such as bead strings [2, 9]. However, a healthy agarwood tree must be physically, chemically, and microbiologically harmed to gradually form agarwood, but the growth is slow and yields are low [10–13]. All species of *Aquilaria* and *Gyrinops* genera have been listed in Appendix II of the convention on international trade in endangered species of wild fauna and flora (CITES) since 2004 [14]. Therefore, the agarwood trade has been closely monitored by international regulators to ensure that such activities do not continue to damage the continued survival of these species in the wild.

Nonetheless, the volume of agarwood traded in the international agarwood trading market reaches several hundred tons each year, with a turnover of tens of millions of dollars. The end market for agarwood is mainly concentrated in Asia and the Middle East [15]. Due to the high demand and high profits in the agarwood market, unscrupulous traders have resorted to counterfeiting and adulteration through various means, such as high-pressure oil injection, soaking spices to pretend to be high-quality agarwood, or using other wood instead of agarwood, especially processing them into handicrafts (beads, bracelets, and statues), this phenomenon is more common [16, 17]. However, the current research on agarwood mainly includes the formation mechanism [18, 19], species identification [20], the separation of sesquiterpenes and 2-(2-phenylethyl) chromones [21], development of modern chromatographic and mass spectrometry analysis methods [22–24], and pharmacology [25, 26]. Although some of the currently developed methods and standards for quality analysis of agarwood can meet the quality analysis and accurate identification of agarwood [27], However, these methods and standards are not suitable for rapid on-site testing of agarwood due to the disadvantages of the large number of samples required, expensive equipment, specialized technicians, and long-term limitations. Liu et al. proposed that the included phloem, also known as interxylary phloem, and rays are the main sites of resin formation in *A. sinensis* [28, 29]. The agar-wit technique was used to induce agarwood formation in *A. sinensis*, which was completely filled with resin the included phloem after one month [30]. Zhang et al. proposed the concept of included phloem, which refers to stranded or striped bundles of abnormal secondary phloem formed by the development of individual vascular formations embedded and dispersed in the secondary xylem of plant roots or stems [31]. Therefore, an understanding of the anatomical structure of agarwood would enable establishing a simple and rapid method for authenticity identification.

Light microscopy is the most widely used method for botanical and timber identification, and the International Association of Wood Anatomists (IAWA) has established a classic database of wood anatomy [32–34]. The presence of resinous included phloem has been reported for *A. sinensis*, *Aquilaria malaccensis* Lam, and *Aquilaria yunnanensis* S.C. Huang [15, 35, 36], but whether or not resinous included phloem is a characteristic structure of agarwood is not clear. For the identification of authentic agarwood, it needs to be confirmed whether the resin in the included phloem is naturally occurring or can be formed by artificially adding extracts. Therefore, it is important to establish a reliable method to identify authentic agarwood. In this study, we validated the resinous included phloem as a key characteristic structure for the authentication of agarwood. This will provide a convenient and rapid method for promoting and popularizing agarwood authentication in trade, customs enforcement, and CITES management.

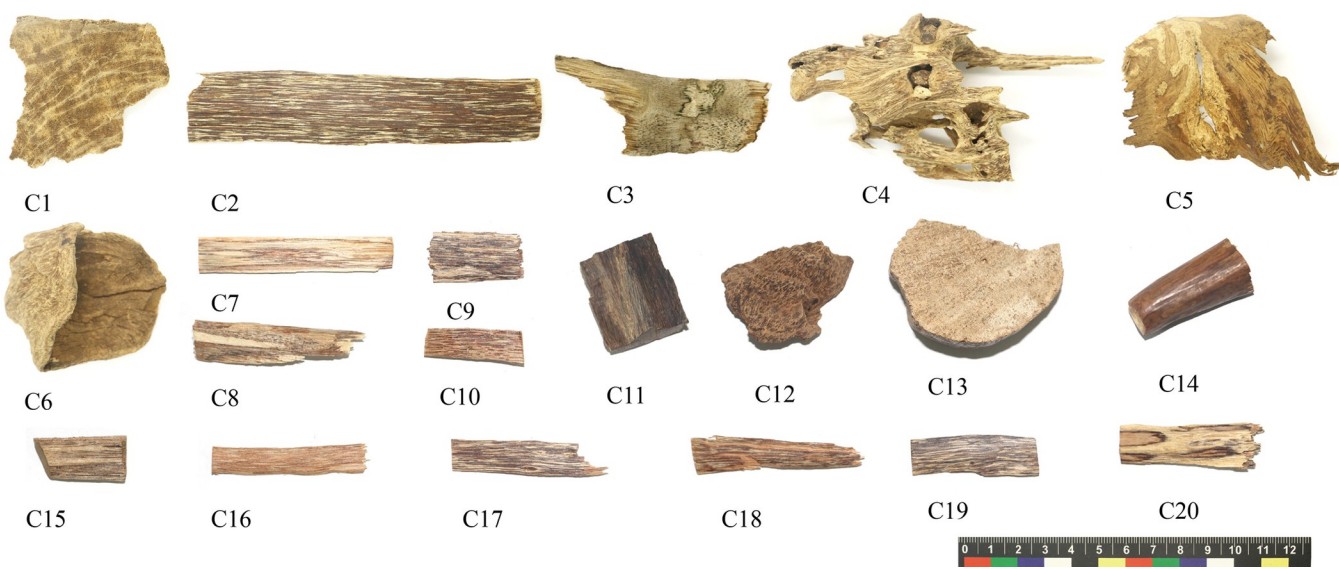

**Fig 1. The experimental agarwood samples from various producing areas.** C1-C20: sample number.

## Materials and methods

### Materials collection

A total of 53 samples were collected in this study (S1 Table). 20 samples (C1-C20) from 17 agarwood-producing areas in 8 countries (China, Bangladesh, Laos, Cambodia, Vietnam, Papua New Guinea, Malaysia, and Indonesia), of which 18 samples had species information (Fig 1). 15 samples (C21-C35) made of fake agarwood from the laboratory, mainly referring to the common counterfeiting methods (steaming with extracts) in the market (Fig 2). 18 samples of agarwood (C36-C53) were collected from the trading market (Fig 3). Such collections are permitted and legal. All voucher specimens of agarwood were kept in the herbarium of the agarwood Identification Center, Hainan Branch, Institute of Medicinal Plants, Chinese Academy of Medical Sciences.

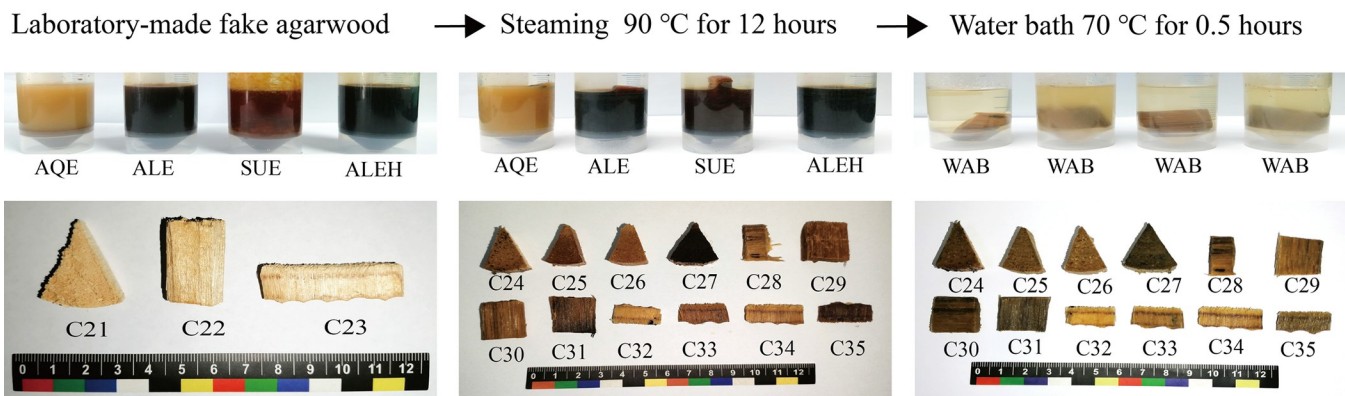

**Fig 2. The experimental agarwood samples from laboratory-made fake agarwood.** C21-C35: sample number. Resin-free agarwood (C21), ordinary agarwood A (C22), and ordinary agarwood B (C23) was soaked in four agarwood extracts and steaming at 90°C for 12 hours in a water bath, followed by 0.5 hours in a 70°C water bath. AQE is an aqueous extract of agarwood. ALE is an alcoholic extract of agarwood. SUE is a supercritical extract of agarwood. ALEH is an alcoholic extract with a high resin content (Chi-Nan) of agarwood. WAB is water.

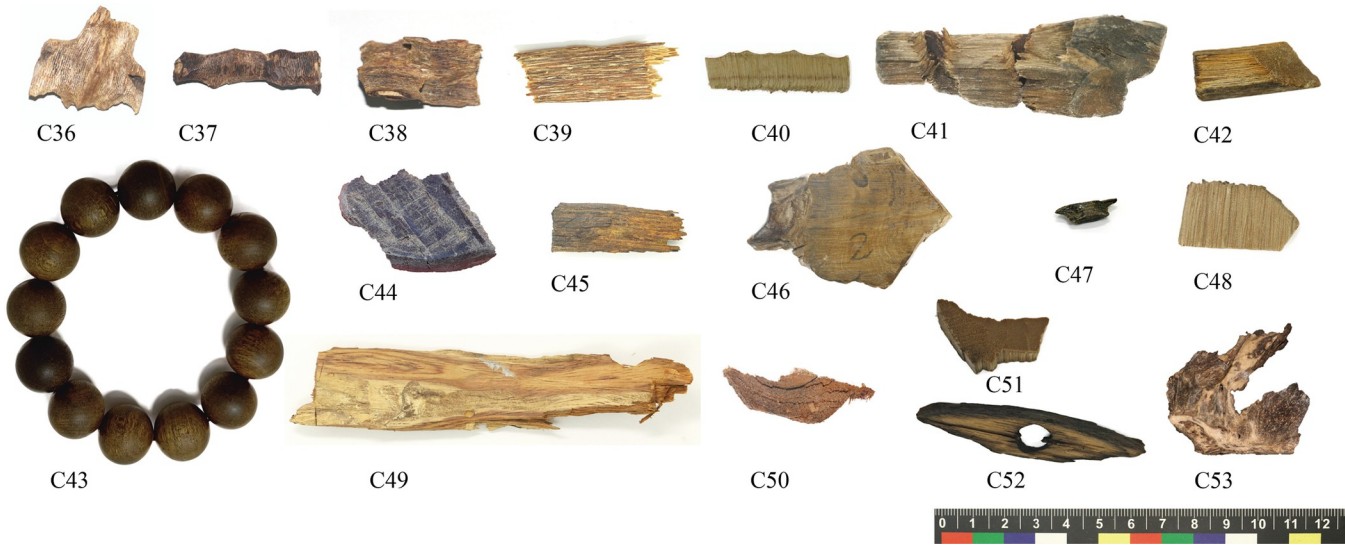

**Fig 3. The experimental agarwood from the market.** C36-C53: sample number.

## Apparatus and reagents

A biological microscope (Nikon Eclipse 80i, Japan) fitted with a camera system (Nikon DS-5Mc, Japan) was used to observe and record the microscopic features of the agarwood samples. The tools included a water bath kettle (HH4, China), tissue softener (glycerin-ethanol, 1:1), chloral hydrate solution (Macklin, China), scalpel, and centrifugal tube.

## Treatment, observation and data analysis

We mimicked the steaming methods that are commonly used on the market to produce fake agarwood. Usually, agarwood extracts are mixed with resin-free agarwood wood or low-quality agarwood and counterfeited by steaming. The main steps of the laboratory-made fake agarwood were to place resin-free agarwood and ordinary agarwood in centrifuge tubes containing aqueous extract solution of agarwood (AQE), alcoholic extract solution of agarwood (ALE), supercritical extract solution of agarwood (SUE), and alcoholic extract solution with high resin content Chi-Nan agarwood (ALEH), respectively, and then steam-firing them for 12 hours at 90°C in a water bath (Fig 2). All samples were divided into small pieces and placed individually into centrifuge tubes filled with water and heated at 70°C for 0.5 hours. Using the freehand sectioning approach, thin slices were cut along the sample transection section. At least 12 thin slices were cut out for each sample. Thin-sectioned samples were placed in chloral hydrate solution and examined with a biological microscope, with 12 thin sections randomly observed for each sample, and a field of view with a magnification of 100 was selected to take photographs and measure the dimensions of the included phloem, the vessel, and the ray of the agarwood samples. The analyzed data were recorded using Excel and included the tangential length and radial width of the included phloem, the tangential and radial width of the vessel, and the width of the rays. The analytical results of each microstructure were plotted using Graphpad software.

## Results

### Microstructural characteristics of authentic agarwood

Twenty samples of authentic agarwood from various producing areas are shown in Fig 1, and the microstructures of the transection section are in Fig 4. The resinous included phloem, and

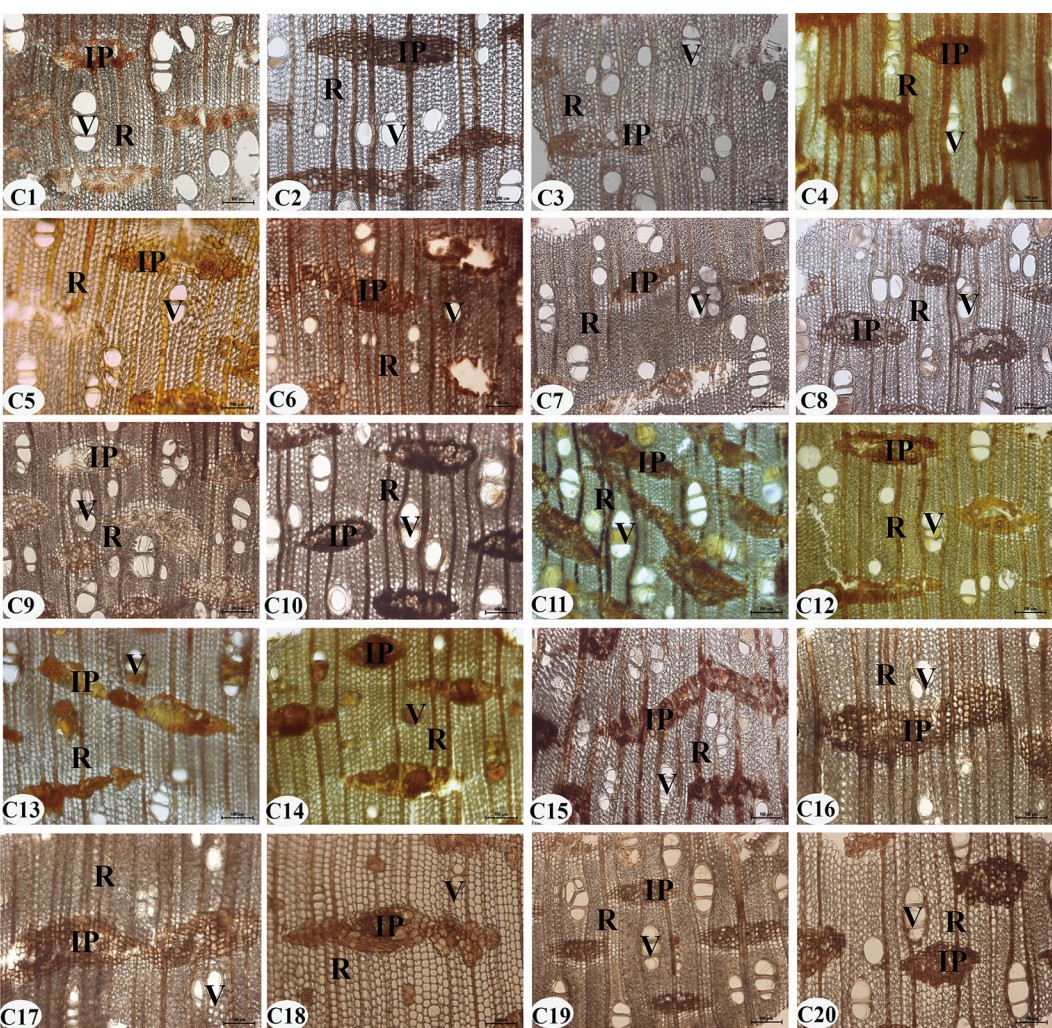

**Fig 4. Microstructure of the transection section of the authentic agarwood from various producing areas.** C1-C20: sample number. IP: included phloem; R: ray; V: vessel. Scale bars 100 μm.

resinous rays and vessels were observed in all samples. The included phloem is diffuse, can be oblong, ribbon-shaped, or island-shaped, and is filled with brown or yellow-brown resin. The tangential and radial widths were 391.63 ± 158.08 (86.98–927.62) μm and 133.49 ± 63.78 (48.09–280) μm, respectively (S1 Fig). Rays are radially extending, exclusively uniseriate, 22.32 ± 9.16 (5.1–62.07) μm wide, and uninterruptedly connected to the included phloem (S1 Fig). Vessels are radially single or multiple, circular or ovoid, with tangential and radial widths of 76.60 ± 28.81 (11.04–223.77) μm and 57.39 ± 24.13 (4.11–162.11) μm, respectively (S1 Fig). Although the size of the included phloem of agarwood is irregular and some samples (e.g., C6/ C7) are not completely filled with resin, it is present in all samples, and it is inferred that the resinous included phloem is a characteristic structure of authentic agarwood.

## Microstructural characteristics of manufactured fake agarwood

Although the resinous included phloem is characteristic structures of agarwood, it has not been verified whether the resins in these structures are naturally occurring or can be formed at a later stage by artificial addition. Therefore, this study used resin-free agarwood and ordinary

Steaming (90°C, 12 hours)

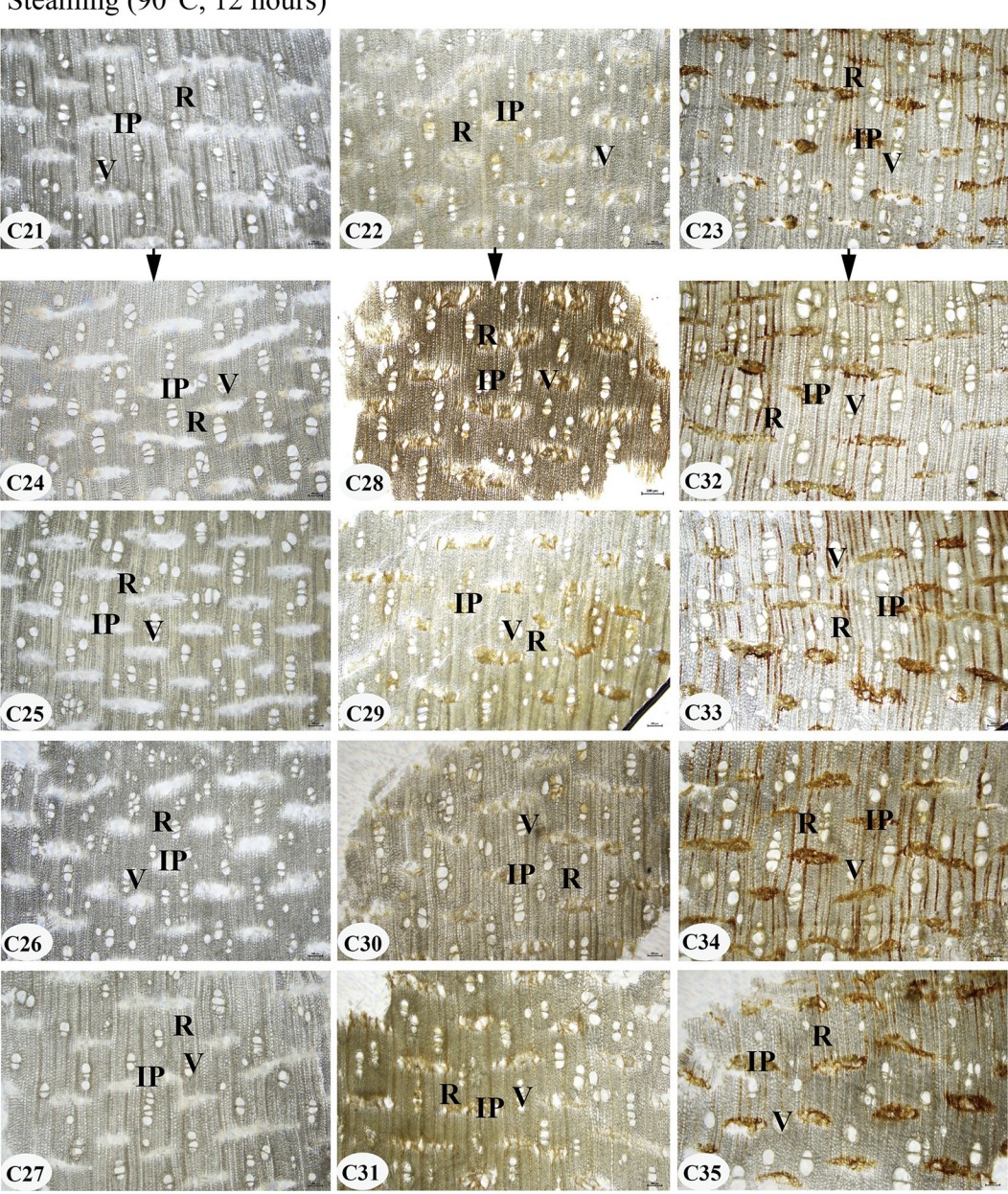

**Fig 5. Microstructures of the transection section of the manufactured fake agarwood steaming 90˚C for 12 hours.**
C21-C35: sample number; IP: included phloem; R: ray; V: vessel. Scale bars 200 μm.

agarwood to simulate the counterfeiting process by steaming the samples with the addition of agarwood extracts and observed the distribution of resin in the included phloem and its difference from the authentic agarwood. After steaming at 90˚C for 12 hours, the samples were all darkened in appearance to varying degrees, as shown in Fig 2. The samples were sectioned, and the microstructures of the included phloem, rays, and vessels were all observed, as shown in Fig 5. Compared to the control samples C21, C22, and C23, the resin did not accumulate efficiently in the included phloem of the samples after treatment with the four agarwood extracts, which was especially observed in the resin-free agarwood

Water bath (70 °C, 0.5 hours)

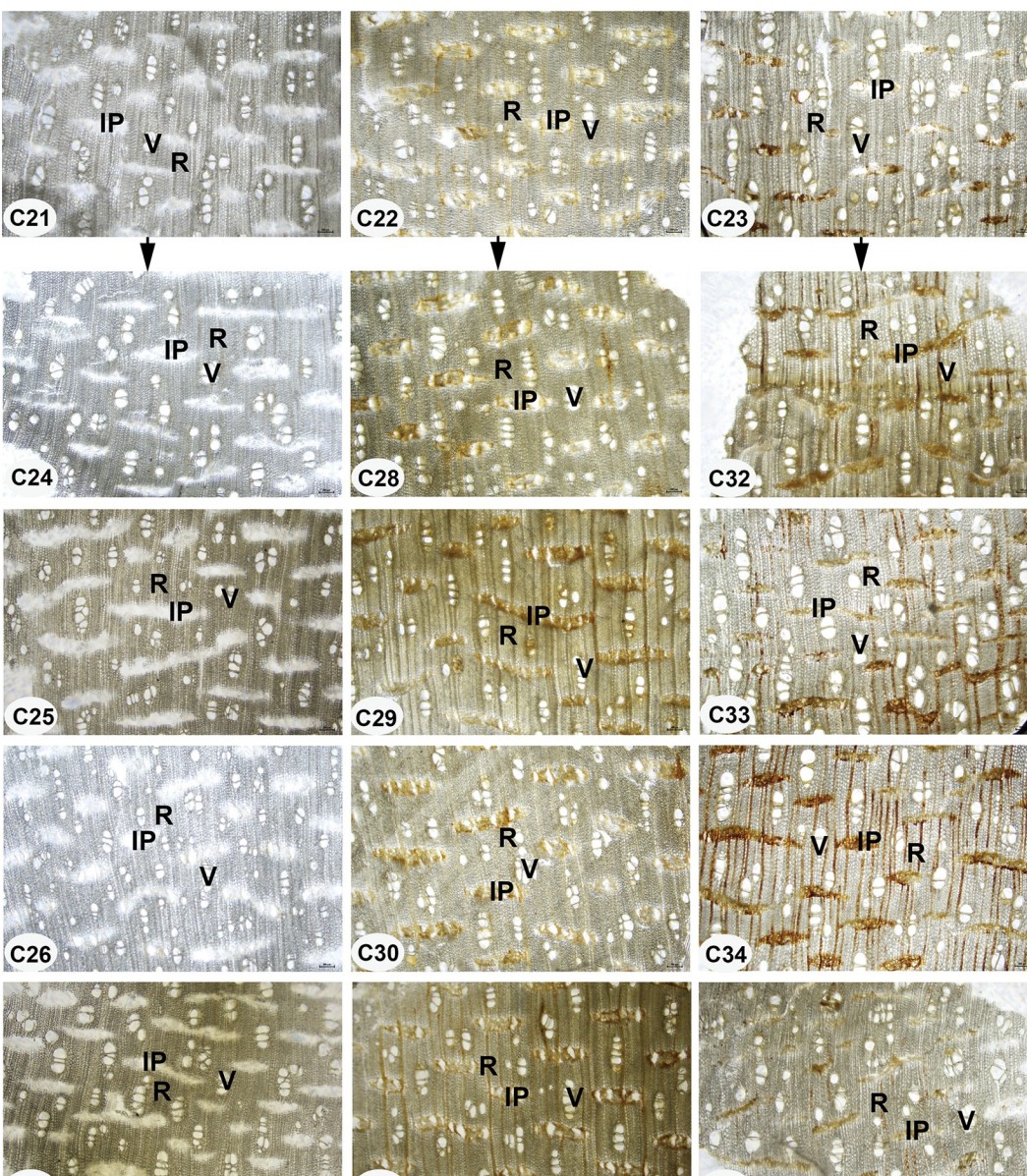

**Fig 6. Microstructures of the transection section of the manufactured fake agarwood after a treatment process followed by a water bath at 70˚C for 0.5 hours.** C21-C35: sample number; IP: included phloem; R: ray; V: vessel. Scale bars 200 μm.

(samples C24, C25, C26, and C27). Meanwhile, the abovementioned samples were further subjected to a water bath at 70˚C for 0.5 h. The samples were further sectioned to observe the microstructure of the samples, as shown in Fig 6. The resin in the included phloem of ordinary agarwood A and B remains stable. Thus, the resin in the included phloem of agarwood is naturally occurring and cannot be effectively accumulated even by the artificial addition of agarwood extracts.

## Validation of microstructural characteristics of agarwood samples from the trading market

We observed the microstructures of 18 agarwood samples from the trading market (Fig 7). After comparison with the authentic agarwood, it was revealed that the microstructures of 10 samples (C36, C37, C38, C39, C40, C42, C47, C51, C52, and C53) were consistent with that of agarwood, and the microstructures of eight samples were not consistent with that of agarwood. Samples C36, C37, C38, C39, and C53 had their included phloem completely filled with resin, while samples C40, C42, C47, C51, and C52 had their included phloem barely filled with resin, with only a small amount of resin dispersed. The microstructure of the remaining eight samples differed from that of the agarwood, although in that they appeared to contain a brown substance. In addition, we soaked the remaining samples in water at room temperature. Interestingly, we found that the color of the soaking solution in the agarwood samples was pale yellow. However, the immersed solution colors of the samples C42 and C47 were yellow and reddish brown, respectively. The color of the immersed solution gradually deepened, accompanied by a pungent odor. Therefore, the resinous included phloem can be utilized to simply identify the authenticity of agarwood.

## Discussion

### The resinous included phloem is a characteristic structure of authentic agarwood

Wood identification is very important in the timber trade, fighting against illegal logging, wood certification, and forensic identification. Microanatomy sometimes provides information hidden by gross anatomy [37]. The xylem of *Aquilaria* is the most important structure, and it provides a place for resin formation and accumulation [35, 36]. The included phloem and rays are interconnected to form a living parenchyma cell network in *A. sinensis* wood [28]. Therefore, verifying the prevalence of resinous included phloem in agarwood plays a decisive role in identifying the authenticity of agarwood. In this study, we found that the microstructures of all authentic agarwood contained included phloem with a natural and regular distribution of brown or yellow-brown resin, although some samples had included phloem of different sizes. Gasson et al. report that all *Aquilaria* and *Gyrinops* are CITES listings and that the woods are anatomically quite similar [37, 38]. To further verify this hypothesis, we searched for the presence of included phloem in published papers related to species of the genera *Aquilaria* and *Gyrinops*, as well as other genera of the Thymolaceae. The search revealed that the wood of 12 species of the genus *Aquilaria* and four species of the genus *Gyrinops* contain included phloem (Table 1). Their microstructures are basically the same, which is consistent with the results of this paper. In addition, the species of the genus *Gonystylus* Teijsm. & Binn., *Aetoxylon* Airy Shaw, and *Phaleria* Jack of the family Thymelaeaceae do not included phloem. However, there is no regularity in the size of the included phloem in different species. The size of the included phloem may be related to the plant species of agarwood, the growing environment, the time and the way of agarwood formation.

It is worth noting that some species have also been reported to contain included phloem. This included phloem have the following characteristics: 1) The included phloem is very narrow, with only two to four cells per cell, such as noted in *Stylidium glandulosum* Salisb. (Stylidiaceae). 2) The included phloem is broad and roughly circular, such as seen in *Strychnos madagascariensis* Poir. (Loganiaceae). 3) The included phloem is crushed or stranded, such as noted in *Salvadora persica* L. (Salvadoraceae); 4) The included phloem is small and numerous in the early stage, with a short lifespan and a gradual decrease in number in the later stage,

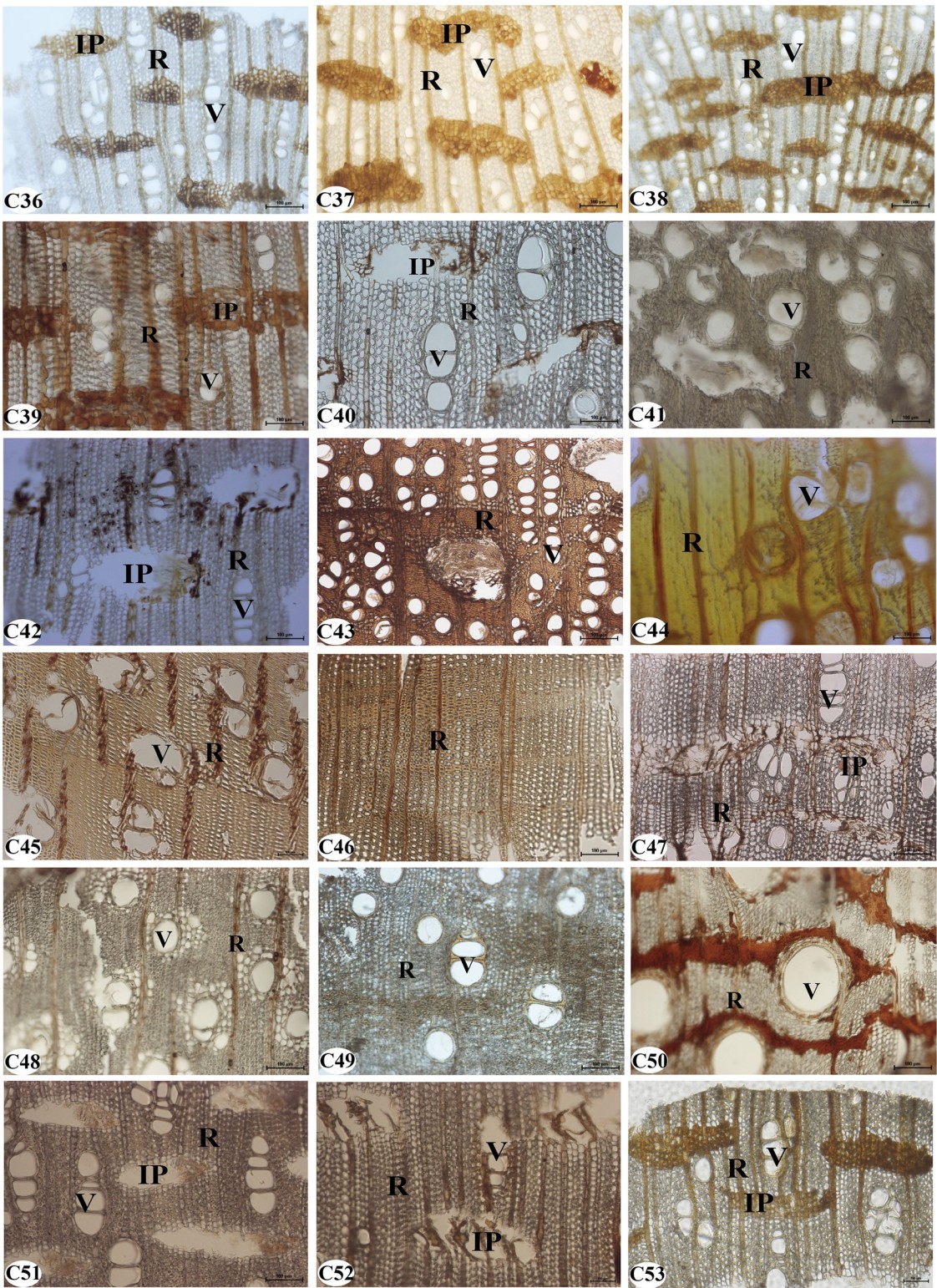

**Fig 7. The microstructure of the transection section of the agarwood samples from the trading market.** C36-C53: sample number. IP: included phloem; R: ray; V: vessel. Scale bars-100 μm.

Table 1. The presence or absence of included phloem structures in different agarwood species.

| Species | Included phloem | References |
|---|---|---|
| *A. agallochum* | + | [40]; This study |
| *A. beccariana* | + | [41] |
| *A. crassna* | + | [42]; This study |
| *A. cumingiana* | + | [41] |
| *A. filaria* | + | This study |
| *A. hirta* | + | [43] |
| *A. khasiana* | + | [44] |
| *A. malaccensis* | + | [36, 42, 43]; This study |
| *A. microcarpa* | + | [43, 45] |
| *A. sinensis* | + | [15, 28, 29, 35, 42, 46]; This study |
| *A. subintegra* | + | This study |
| *A. yunnanensis* | + | [47]; This study |
| *Gyrinops caudata* | + | [41] |
| *G. moluccana* | + | [41] |
| *G. versteegii* | + | [42, 48, 49] |
| *G. walla* | + | [50] |
| *Gonystylus affinis* | - | [10] |
| *G. bancanus* | - | [10] |
| *G. brunnescens* | - | [10] |
| *G.confuses* | - | [10] |
| *G. punctatus* | - | [37] |
| *G. macrophyllus* | - | [51] |
| *Aetoxylon sympetalum* | - | [52] |
| *Pbaleria* sp. | - | [41] |

Note: + Included phloem exist.—Included phloem does not exist.

such as occurs in *Oenothera linifolia* Nutt. (Onagraceae) [39]. However, the microstructure of the included phloem of these reported species is completely different from that of agarwood. Despite the samples similarity in appearance to agarwood, the anatomical methods utilized in this paper allow for the rapid identification of fake agarwood.

## The resin in the included phloem of agarwood is naturally occurring and cannot be effectively accumulated by artificial additions

Agarwood is formed when the agarwood trees are damaged by physical force, insect, or fungal infection, and the resinous substance is embedded at the wound and gradually accumulates over time [10–13]. In our study, the fake samples were dark or black in color due to the counterfeiting of agarwood by the steaming method, and their appearance and smell were similar to those of authentic agarwood. However, by microscopic observation, the resin in the included phloem of these fake agarwood samples (resin-free agarwood wood and ordinary agarwood) did not accumulate effectively. It has been reported that the addition of illegal additives (carbon powder, rosin, extracts, colored oil, kerosene, etc.) can be used to increase the wood's weight and color and create a appearance resemblance to high-quality agarwood [1].

In general, the included phloem of authentic agarwood are supposed to contain resin. The results of this study revealed that the resin in the included phloem of some of the agarwood samples were incompletely filled, a phenomenon that may be related to the short time of

agarwood formation, the unstable method of induction, or incomplete primary processing. It has also been shown that when agarwood is immersed in water, the resin dissolves very slowly, and the resin remains stable for a certain period of time [53]. Our validation experiments also confirmed that there was no significant loss or reduction of resin in the included phloem of the authentic agarwood under the same steaming conditions. In addition, we found that the color of the soaking water solution in the authentic agarwood samples stayed pale yellow all the time. However, the soaking water solution colors of 2 samples from the trading market were yellow and reddish brown, respectively. The color of the immersed solution gradually deepened, accompanied by a pungent odor. Therefore, we conclude that additives have been added to the agarwood.

## Searching for resinous included phloem under the microscope provides a convenient and rapid method for the authentication of agarwood

We selected included phloem as important elements to identify in all samples of agarwood. The method was validated by identifying the authentic agarwood sourced from the market, as well as fake agarwood. We have formed this method of authentication of agarwood into a standard operation procedure, as shown in Fig 8. which first described the appearance trait of the samples, followed by basic sample processing, including sampling, warm bath, permeabilization, and sealing; finally, the included phloem containing resin were observed under the microscope to determine the authenticity of the samples. Microstructure identification

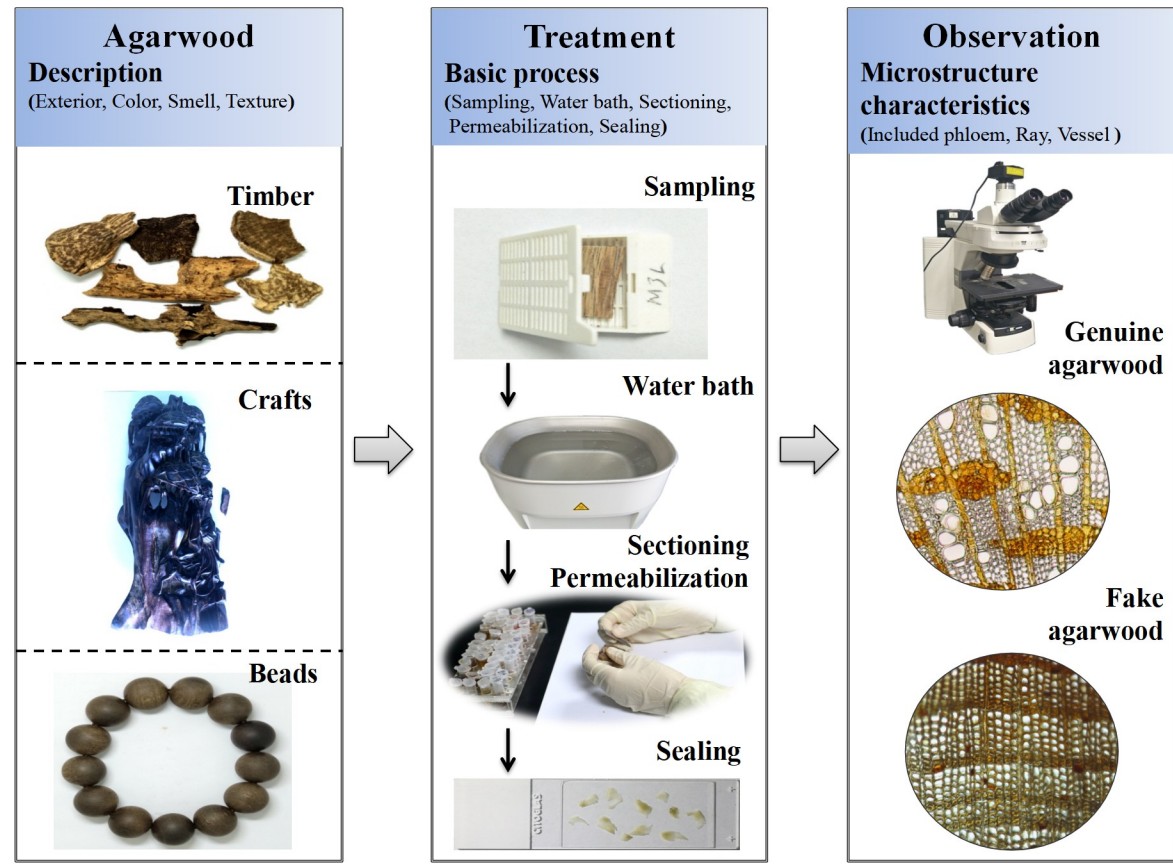

**Fig 8. Standard operating procedures for microstructure identification of agarwood.**

**Table 2. Comparison of the results of this study with existing techniques for the identification of agarwood.**

| Item | Microstructure | Traits | Chemical analysis | Molecular identification |
|---|---|---|---|---|
| Equipment | Microscope | Eyes, hands, and nose | Chromatograph, mass spectrometer, and its combination instrument | PCR amplifier, and DNA Sequencers |
| Detection time | 1–2 hours | Less than 0.5 hour | 5–8 hours | More than 10 hours |
| Cost | $7 | Uncertainty | $120 | $140 |
| Technician | Not required | Required | Required | Required |
| Result reliability | Reliable; The internal structure and resins of the agarwood to be measured can be observed. | Unreliable; It requires a great deal of experience and judgment. | It can identify the quality of incense but cannot determine whether it is fake. | It can identify the species of agarwood but cannotdetermine whether it is fake. |

provides reliable results compared to the trait identification of agarwood. In addition, compared with chemical analysis and molecular identification, the microstructure identification method is simple, low cost, and does not require expensive and sophisticated equipment. The microstructure identification method used in this study was evaluated in terms of equipment, testing time, cost, technicians, and reliability of results and it was able to identify the authenticity of agarwood quickly, conveniently, and accurately (Table 2).

## Conclusion

In this study, by comparing the authenticity of the microstructures of agarwood from different producing areas and artificially simulated fake agarwood, it was found that the included phloem of authentic agarwood was filled with brown to brownish resins, which were also deposited in the rays, naturally distributed, and stably present. The filling of resin in the included phloem of agarwood did not occur after artificially simulated counterfeiting treatments. Fake agarwood is divided into two categories, one itself is non-agarwood wood (without included phloem), and the other is artificially counterfeiting (there is included phloem, but it does not contain resin, or counterfeiting filler material can be dissolved out). Therefore, the resinous included phloem is an important characteristic structure of agarwood and can be used as a basis for authenticity identification. This will provide a convenient and rapid method for the promotion and popularization of agarwood authentication in trade, customs enforcement, CITES management, and other fields.

## Supporting information

**S1 Fig. The sizes of authentic agarwood included phloem, ray, and vessel.** A and B: the tangential and radial widths of the included phloem; C: ray width; D and E: the tangential and radial widths of the vessel.
(TIF)

**S1 Table. Information on the samples used in this study.** [#] Wild agarwood, [##] Artificially induced agarwood.; a) aqueous extract solution of agarwood (AQE), b) alcoholic extract solution of agarwood (ALE), c) supercritical extract solution of agarwood (SUE), d) alcoholic extract solution with high resin content Chi-Nan agarwood (ALEH).
(DOCX)

## Acknowledgments

We thank the agarwood Identification Center, Hainan Branch, Institute of Medicinal Plants, Chinese Academy of Medical Sciences and Peking Union Medical College for supporting the collection of samples and microstructure experiments in this study.

## Author Contributions

**Conceptualization:** Yangyang Liu, Jianhe Wei.

**Data curation:** Jian Feng, Anzhen Xie.

**Formal analysis:** Jian Feng.

**Investigation:** Jian Feng, Peiwei Liu, Yun Yang.

**Methodology:** Jian Feng, Yangyang Liu, Peiwei Liu, Jianhe Wei.

**Resources:** Jian Feng, Yangyang Liu, Jianhe Wei.

**Validation:** Jian Feng, Peiwei Liu, Anzhen Xie.

**Visualization:** Jian Feng.

**Writing – original draft:** Jian Feng.

**Writing – review & editing:** Jian Feng, Yangyang Liu, Jianhe Wei.

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
