## [Decision Letter · Decision Letter 0]

23 Aug 2024

PONE-D-24-12796The presence of resinous included phloem indicative of genuine or fake agarwoodPLOS ONE

Dear Dr. Liu,

Thank you for submitting your manuscript to PLOS ONE. The independent reviewers have highlighted few important shortcomings in your manuscript. Please go through the comments and submit your revised manuscript.

We look forward to receiving your revised manuscript.

Kind regards,

Pankaj Bhardwaj, Ph.D.

Academic Editor

PLOS ONE

“Hainan Provincial Natural Science Foundation (822MS145); the project of medical and health science and technology innovation engineering of Chinese Academy of Medical Sciences (2021-1-I2M-032);Hainan Provincial Nanhai xinxing Science and Technology Innovation Talent Platform Project (NHXXRCXM202341); Hainan Provincial Nanhai xinxing Science and Technology Innovation Talent Platform Project (NHXXRCXM202341); National Natural Science Foundation of China (Grant No.81703660); the earmarked fund for CARS (CARS-21).”

Reviewers' comments:

Reviewer's Responses to Questions

**Comments to the Author**

1. Is the manuscript technically sound, and do the data support the conclusions?

Reviewer #1: Yes

Reviewer #2: Partly

2. Has the statistical analysis been performed appropriately and rigorously? 

Reviewer #1: I Don't Know

Reviewer #2: No

3. Have the authors made all data underlying the findings in their manuscript fully available?

Reviewer #1: No

Reviewer #2: No

4. Is the manuscript presented in an intelligible fashion and written in standard English?

Reviewer #1: No

Reviewer #2: No

5. Review Comments to the Author

Reviewer #1: This work describes the method for identifying authentic agarwood.

General observations:

1. The language needs improvement.

2. The supplementary information can form a part of the main text.

3. Data related to the anatomical characteristics and be provided as supplementary information.

The sectionwise comments are provided as attachment.

Reviewer #2: 1. The manuscript is very flimsy in its scientific content and poor in language. It needs a major revision before being considered further for evaluation.

2. The authors have included many paragraphs that are irrelevant; it seems like they were included just to increase the volume of the text.

3. The details of any statistical analysis conducted are not clear.

4. Is the repeatability of the experiments doubtful?

5. The title of the manuscript is inappropriate and does not spell out the objective of the study clearly.

6. The manuscript contains many typographical errors.

7. The plagiarism of the manuscript may also be checked.

6. PLOS authors have the option to publish the peer review history of their article (what does this mean?). If published, this will include your full peer review and any attached files.

Reviewer #1: No

Reviewer #2: **Yes: **SHARADA MALLUBHOTLA

---

## [Author Response · Author response to Decision Letter 0]

23 Sep 2024

Point-by-point response to reviewer 1

Reply to journal requirements

We are very grateful to the reviewer for the specification of this paper, and we have made the required article-by-article changes as detailed below:

Q1. The language needs improvement.

Answer: Thank you very much for your review. The language of the manuscript has been revised and embellished.

Q2. The supplementary information can form a part of the main text.

Answer: Thank you very much for your review. Changes have been made in response to your comments, and S1a Fig, S1b Fig, S1c Fig, and S1 Table have been added to the manuscript. To avoid duplication, Table 1 has been placed in the annexed material as S1 Table. Revisions to other elements have been identified in the manuscript with yellow underlining.

Q3. Data related to the anatomical characteristics and be provided as supplementary information. The sectionwise comments are provided as attachment.

Answer: Thank you very much for your review. In response to your comments, we have made some changes and have provided the data charts for anatomical features as an attachment, and other content changes are identified with a yellow underline.

Q4. Title: Could be modified as 1) Assessing Authenticity in Agarwood through the Presence of Resinous Included Phloem; 2) Resinous Included Phloem as a Key Indicator of fake or authentic Agarwood.

Answer: Thank you very much for your review. We have chosen your point 2 as a more appropriate title for the article and have revised it in the manuscript.

Q5. Introduction: 1) Language needs improvement; 2) A logical flow is lacking in the introduction. It fails to introduce the need for identifying included phloem characterisitics to establish a reliable method for identifying authentic agarwood. 

Answer: Thank you very much for your review. We have made changes to the introduction.

Q6. Materials and Methods: 1) Clarify the terms "uncertain agarwood", " producing areas " to ensure it's easily understood by readers. 2) What does ‘mainly referring to the common counterfeiting methods in the market’ mean? 3) When you mention 53 batches, how many samples were present in each batch? Or do you mean 53 samples? 4) Ensure consistency in terminology. Retain either batch or sample. 5) How many observations were recorded per sample? 6) What statistical analysis was employed? 7) Overall, the essential details of the sample collection process is conveyed, but breaking down the information and providing clarifications can enhance readability and comprehension. 

Answer: Thank you very much for your review. We have revised and clarified the seven issues you raised in the manuscript.

Q7. Results: The results effectively communicate the findings, but there are areas where clarity and conciseness could be improved.

Answer: Thank you very much for your review. We have followed the changes made to the results section.

Q8. Discussion: 1) The readability of the discussion is moderate. While the information presented is clear in most parts, there are areas where the complexity of the subject matter or the improper usage of grammar poses a challenge. 2) With minor revisions to simplify the language and complex concepts, the readability of the report could be improved. 3) Consider rephrasing or condensing repetitive information to avoid redundancy.

Answer: Thank you very much for your review. We have made the changes you suggested.

Point-by-point response to reviewer 2

Reply to journal requirements

We are very grateful to reviewer for the specification of this paper, and we have made the required article-by-article changes as detailed below:

Q1. The manuscript is very flimsy in its scientific content and poor in language. It needs a major revision before being considered further for evaluation.

Answer: Thank you very much for your review. The language and content of the manuscript has been revised and embellished.

Q2. The authors have included many paragraphs that are irrelevant; it seems like they were included just to increase the volume of the text.

Answer: Thank you very much for your review. The contents of the manuscript have been verified and revised.

Q3. The details of any statistical analysis conducted are not clear.

Answer: Thank you very much for your review. Sample observation and statistical analysis methods have been added to the Materials and Methods section.

Q4. Is the repeatability of the experiments doubtful?

Answer: Thank you very much for your review. We firmly believe that the results of the manuscript are reliable. We analyzed the samples of authentic agarwood from different producing areas by observation, and at least 12 replicates of thin slices of each sample were extracted for microstructure observation and analysis. Methodological validation was also carried out on artificially fake agarwood, and the resin-containing included phloem was found to be the characteristic structure of authentic agarwood. In addition, our team specializes in the research of authentication and quality control of agarwood and has established the China's first agarwood authentication center, which is also confirmed by the CMA accreditation issued by the Chinese government. The methods used in this research are simple and practical and meet the use of agarwood-related industries.

Q5. The title of the manuscript is inappropriate and does not spell out the objective of the study clearly.

Answer: Thank you very much for your review. We chose the appropriate title “Resinous included phloem as a key indicator of authentic or fake agarwood” and revised it in the original manuscript.

Q6. The manuscript contains many typographical errors.

Answer: Thank you very much for your review. Changes have been checked and corrected in the manuscript.

Q7. The plagiarism of the manuscript may also be checked.

Answer: Thank you very much for your review. The data in our manuscript is authentic and reliable and there is no plagiarism. our team specializes in the research of authentication and quality control of agarwood and has established the China's first agarwood authentication center, which is also confirmed by the CMA accreditation issued by the Chinese government. The methods used in this research are simple and practical and meet the use of agarwood-related industries. The experimental data in the manuscripts involved are authentic.

---

## [Decision Letter · Decision Letter 1]

2 Oct 2024

Resinous included phloem as a key indicator of authentic or fake agarwood

PONE-D-24-12796R1

Dear Dr. Liu,

We’re pleased to inform you that your manuscript has been judged scientifically suitable for publication and will be formally accepted for publication once it meets all outstanding technical requirements.

Kind regards,

Pankaj Bhardwaj, Ph.D.

Academic Editor

PLOS ONE

Additional Editor Comments (optional):

Reviewers' comments:

Reviewer's Responses to Questions

**Comments to the Author**

1. If the authors have adequately addressed your comments raised in a previous round of review and you feel that this manuscript is now acceptable for publication, you may indicate that here to bypass the “Comments to the Author” section, enter your conflict of interest statement in the “Confidential to Editor” section, and submit your "Accept" recommendation.

Reviewer #1: All comments have been addressed

Reviewer #2: All comments have been addressed

2. Is the manuscript technically sound, and do the data support the conclusions?

Reviewer #1: Yes

Reviewer #2: Yes

3. Has the statistical analysis been performed appropriately and rigorously? 

Reviewer #1: Yes

Reviewer #2: Yes

4. Have the authors made all data underlying the findings in their manuscript fully available?

Reviewer #1: Yes

Reviewer #2: Yes

5. Is the manuscript presented in an intelligible fashion and written in standard English?

Reviewer #1: Yes

Reviewer #2: Yes

6. Review Comments to the Author

Reviewer #1: All corrections / suggestions in the article - The presence of resinous included phloem indicative of genuine or fake agarwood have been addressed. The article can be accepted.

Reviewer #2: The authors have addressed the comments to my satisfaction.

The statistial analysis an important component of practical research has been substantially added to the manuscript.

The manuscript can now be accepted after formatting the references as per journal guidelines.

7. PLOS authors have the option to publish the peer review history of their article (what does this mean?). If published, this will include your full peer review and any attached files.

Reviewer #1: No

Reviewer #2: **Yes: **SHARADA MALLUBHOTLA

---

## [Editor Report · Acceptance letter]

16 Oct 2024

PONE-D-24-12796R1 

PLOS ONE

Dear Dr. Liu, 

I'm pleased to inform you that your manuscript has been deemed suitable for publication in PLOS ONE. Congratulations! Your manuscript is now being handed over to our production team.

Kind regards, 

on behalf of

Dr. Pankaj Bhardwaj 

Academic Editor

PLOS ONE